# Microneurotrophin BNN27 Reduces Astrogliosis and Increases Density of Neurons and Implanted Neural Stem Cell-Derived Cells after Spinal Cord Injury

**DOI:** 10.3390/biomedicines11041170

**Published:** 2023-04-13

**Authors:** Konstantina Georgelou, Erasmia-Angeliki Saridaki, Kanelina Karali, Argyri Papagiannaki, Ioannis Charalampopoulos, Achille Gravanis, Dimitrios S. Tzeranis

**Affiliations:** 1Department of Pharmacology, School of Medicine, University of Crete, 71003 Heraklion, Greece; 2Institute of Molecular Biology and Biotechnology, Foundation for Research and Technology-Hellas, 71003 Heraklion, Greece; 3Department of Mechanical and Manufacturing Engineering, University of Cyprus, Nicosia 2109, Cyprus

**Keywords:** spinal cord injury, neurotrophins, astrogliosis, neuroprotection, neural stem cells, biomaterials

## Abstract

Microneurotrophins, small-molecule mimetics of endogenous neurotrophins, have demonstrated significant therapeutic effects on various animal models of neurological diseases. Nevertheless, their effects on central nervous system injuries remain unknown. Herein, we evaluate the effects of microneurotrophin BNN27, an NGF analog, in the mouse dorsal column crush spinal cord injury (SCI) model. BNN27 was delivered systemically either by itself or combined with neural stem cell (NSC)-seeded collagen-based scaffold grafts, demonstrated recently to improve locomotion performance in the same SCI model. Data validate the ability of NSC-seeded grafts to enhance locomotion recovery, neuronal cell integration with surrounding tissues, axonal elongation and angiogenesis. Our findings also show that systemic administration of BNN27 significantly reduced astrogliosis and increased neuron density in mice SCI lesion sites at 12 weeks post injury. Furthermore, when BNN27 administration was combined with NSC-seeded PCS grafts, BNN27 increased the density of survived implanted NSC-derived cells, possibly addressing a major challenge of NSC-based SCI treatments. In conclusion, this study provides evidence that small-molecule mimetics of endogenous neurotrophins can contribute to effective combinatorial treatments for SCI, by simultaneously regulating key events of SCI and supporting grafted cell therapies in the lesion site.

## 1. Introduction

Spinal cord injury (SCI) leads to partial or complete loss of sensory and/or motor functions. According to the World Health Organization, approximately 400,000 individuals around the world suffer a SCI each year. Despite significant efforts, existing clinical treatments are limited to surgical intervention for spinal cord decompression and the controversial administration of methylprednisolone, a corticosteroid that has serious adverse effects and poor efficacy [1]. The development of effective SCI treatments remains an unmet clinical need.

SCI triggers a complex cascade of interrelated events. Acute primary injury (cell death, hemorrhage, edema and ischemia) proceeds to secondary injury (inflammatory and cytotoxic events) within the next few hours to weeks and leads to chronic injury over the next months to years [2]. The multifactorial nature of SCI calls for the development of combinatorial therapies that will combine the complementary effects of multiple components to simultaneously regulate multiple events.

Significant research focuses on developing combinatorial treatments for SCI that integrate the complementary actions of biomaterials, cell therapies and diffusible growth factors [3]. In such combinatorial treatments, biomaterials contribute structural support, inflammation regulation, delivery, protection and orientation of cells, a substrate for cellular migration and axonal elongation, and localized delivery of diffusible therapeutic molecules (small molecules, biologics) at the lesion [1]. The cellular part of combinatorial treatments usually focuses on replacing neural cells lost during injury and re-establishing connectivity in the lesion site [4]. Among the various kinds of stem cells tested in experimental SCI treatments, neural stem cells (NSCs) have shown promising results in animal models, including the extension of a large number of axons caudally to the lesion and synapse creation with corticospinal tract axons [5,6]. Despite effort to translate NSC-based treatments towards human clinical evaluation [7], much work remains to enhance their efficacy. In this direction, much research focuses on optimizing the biomaterial part of SCI treatments and complementing grafted cells with soluble factors. Recently, we demonstrated that porous collagen scaffold (PCS) grafts can safely deliver NSCs in SCI lesions, enabling their differentiation into neurons and glia. Such NSC-seeded PCS grafts significantly improved locomotion recovery over a period of 12 weeks in a mouse dorsal column crush SCI model [8]. This work focuses on the third part of combinatorial SCI treatments, the supplementation of cell therapies with soluble therapeutic molecules, emphasizing neurotrophic factors.

Much research on SCI treatments has focused on trophic factors’ support, due to their role in neuronal survival, axonal guidance and synapse formation [9,10]. Among trophic factors, NGF, the prototypic member of the neurotrophin family, has received significant attention. NGF and its receptors (TrkA, p75^NTR^) are expressed in the spinal cord [11,12,13]. The poor regenerative capacity of the CNS compared to the peripheral nervous system (PNS) has been partly attributed to inadequate trophic support, including low levels of NGF following injury. Indeed, following SCI, NGF expression in the spinal cord was detected just in meningeal cells and Schwann cells in nerve roots [9]. SCI does not induce NGF expression in spinal cord oligodendrocytes, in contrast to the PNS axotomy that induces robust NGF expression in Schwann cells. These observations suggest trophic factor supplementation as a potential therapeutic strategy for SCI. Following SCI, NGF administration induced tract-specific neurite growth [14,15], reduced neuronal death [16,17,18], inhibited SCI-induced autophagy [18], and led to behavioral improvement [16,17,18,19]. Significant research has focused on developing means of NGF administration that bypass the poor penetration of NGF via the blood–brain barrier (BBB) [20], including intranasal delivery [13], local BBB disruption using ultrasound [17] and the delivery of NGF-expressing genetically modified cells, such as fibroblasts [14,21,22], Schwann cells [23] or NSCs [18,19], in the lesion site. Overall, despite some promising results, the utilization of NGF in combinatorial research treatments for SCI has been limited, due to unfavorable pharmacokinetics and side effects, including hyperalgesia [24].

Key limitations of endogenous neurotrophins regarding therapeutic applications can be bypassed by microneurotrophins (MNT), 17-carbon small-molecule derivatives of dehydroepiandrosterone that specifically activate neurotrophin receptors and can pass the BBB [25]. BNN27, the seminal ΜΝΤ, is an NGF analog that can activate both its receptors (TrkA, p75^NTR^) and lacks the hyperalgesia effects of NGF. BNN27 has demonstrated various effects related to wound healing: BNN27 demonstrated neuroprotective effects on various kinds of neurons (superior cervical ganglion cells, dorsal root ganglion cells, cerebellar granule neurons) mediated via both NGF receptors and downstream pro-survival pathways [26,27], decreased apoptosis of TrkA^+^ sensory neurons in E13.5 NGF null mice embryos [26], decreased retina cell apoptosis and astrogliosis in a rat model of diabetes-induced retinal damage [28,29], and reduced microgliosis in a mouse cuprizone-induced demyelination model [30]. Nevertheless, the effects of BNN27 have not been evaluated yet in any CNS injury model, either in the form of a monotherapy or as a part of a combinatorial therapy.

The present study evaluates the effects of BNN27 in the mouse dorsal column crush SCI model. BNN27 was delivered systemically either by itself or combined with NSC-seeded PCS grafts demonstrated recently to improve locomotion performance in the same SCI model [8]. Our findings highlight the central role of NSC-seeded PCS grafts in locomotion improvement, the replenishment of neural cells lost due to SCI and the integration of neuronal cells with surrounding CNS tissue 12 weeks post injury (wpi). While systemic administration of BNN27 did not improve locomotion recovery over a 10-week period following SCI, BNN27 significantly reduced astrogliosis and increased neuron density in the lesion site at 12 wpi. Furthermore, BNN27 increased the density of implanted NSC-derived cells in the lesion. Our study provides evidence that small-molecule mimetics of endogenous neurotrophins can contribute to effective combinatorial SCI treatments by regulating key events of SCI and supporting grafted cell therapies in the lesion site.

## 2. Materials and Methods

### 2.1. Animals

Animal care and experimentation protocols were performed according to the approval of the Veterinary Directorate of the region of Crete in compliance with Greek Government guidelines, EU guidelines 2010/63/EU, FORTH ethics committee guidelines and in accordance with approved protocols from the Federation of European Laboratory Animal Science Associations (FELASA) on use of laboratory animals. Facility License number: EL91-BIOexp-02; Project title: “Investigating the effects of neurotrophins and their synthetic analogues on the mechanisms of neurodegeneration and neurogenesis in Alzheimer’s disease and the effect of neuroimplants based on porous collagen scaffolds in the central nervous system trauma”; approval numbers: 262,272 and 360,667; date of initial approval: 29 October 2018; date of reapproval: 29 November 2021. C57/BL6 [31] and Rosa26-YFP mice (containing a floxed yellow fluorescent protein (YFP) gene) [32] were maintained in climate-controlled conditions (30–50% humidity, 21  ±  2  °C, 12:12  h light/dark cycle) on a 12 h light/dark cycle with ad libitum access to food and water. Rosa26-YFP mice were crossed with CMV-Cre transgenic mice [33] to obtain mice that expressed YFP in all tissues.

### 2.2. Scaffold Fabrication

PCS sheets were fabricated by lyophilizing a 5 mg/mL microfibrillar collagen I suspension in 50 mM acetic acid as described previously [8]. The resulting 2.5 mm-thick dry sheets were cross-linked via dehydro-thermal treatment (105 °C, 50 mTorr, 24 h). Scaffold structure was verified by scanning electron microscopy.

### 2.3. Primary Neural Stem Cell Isolation and Culture

NSC isolation was performed from brain tissue (cortex) of mouse embryos (E13.5) of Rosa26-YFP mice crossed with CMV-Cre transgenic mice via the enzymatic digestion of the cortex. Specifically, pregnant mice (gestational day 13.5) were sacrificed via cervical dislocation and embryos were carefully removed. Embryos’ cortical hemispheres were washed gently in Hanks’ balanced salt solution (Thermo scientific) with 5% penicillin/streptomycin and were mechanically dissociated for NSC isolation in complete NSC medium (Dulbecco’s modified Eagle’s medium/F12 (Thermo scientific), B27 supplement minus vitamin A (Thermo scientific), 0.6% D-glucose, 50 mg/mL primocin (InvivoGen), 20 μg/mL fibroblast growth factor 2 (R&D), 20 μg/mL epidermal growth factor (R&D)). Then, 2.5∙10^5^ NSCs were seeded in T25 flasks in 5 mL complete NSC medium and 1 mL of fresh medium was added every other day. Neurospheres formed within 2 days and were dissociated by Accutase (Sigma) at day 4 or 5. Dissociated NSCs were either passaged or used for experiments (passage 3–8). Care was taken in the first passage to pick and keep neurospheres of YFP-expressing embryos.

### 2.4. Spinal Cord Injury Animal Model

Procedures for SCI and graft implantation followed previously published protocols [8]. Specifically, 3 days prior to injury, 3∙10^4^ YFP-expressing NSCs were seeded into 1 × 1 × 1.5 mm PCS samples and cultured in complete NSC medium. Prior to implantation, grafts were washed for 30 min in phosphate-buffered saline (PBS). All surfaces, tools, and instruments utilized had been sterilized carefully. C57/BL6 male mice approximately 4 weeks old were anesthetized by 2.5:1 isoflurane:oxygen mix inhalation for 5 min in a scavenger box until breathing slowed down. Before anesthesia, a drop of meloxicam (Metacam) was introduced per os. After verifying the absence of paw reflexes, each mouse was shaved in the level of the humpback and the exposed skin was disinfected. Each mouse was then transferred onto a heat pad, where ophthalmic ointment was applied to avoid eye dryness and 2:1 isoflurane:oxygen mix was applied to maintain deep anesthesia via a mask. In all animal groups (“Uninjured control”, “Placebo”, “BNN27”, “Scaffold NSC + Placebo”, “Scaffold NSC + BNN27”) a skin incision was made from the base of the humpback until the higher point of the rib cage and exposed muscles were carefully torn. After removing surrounding ligaments, the bone of T10 vertebra was removed and an incision was made in the dura matter above the T13 segment [34]. In four injured animal groups (“Placebo”, “BNN27”, “Scaffold NSC + Placebo”, “Scaffold NSC + BNN27”) the dorsal column was crushed by inserting Dumont #5 fine forceps 1 mm deep into the white matter and keeping them closed for 10 sec. This step was repeated once to create a ≈1 mm^3^ pocket (lesion). In two animal groups (“Scaffold NSC + Placebo”, “Scaffold NSC + BNN27”) a NSC-seeded graft was placed in the pocket immediately following injury. In all animal groups, the exposed spinal cord tissue was covered by a Geistlich membrane (Geistlich Bio-Gide) and then a hemostatic agent (Lyostypt, B. Braun Company), the muscles and skin were sutured and the skin was disinfected. Then, in animals of the “BNN27” and “Scaffold NSC + BNN27” groups, a small incision was made close to the shoulder and a pellet that contained 18 mg BNN27 (60 day release period; Innovative Research of America #SX999) was implanted under the skin. After inserting the pellet, the skin was sutured and disinfected. Similarly, in animals of the “Placebo” and “Scaffold NSC + Placebo” groups, mice were implanted with a pellet that lacked an active pharmaceutical ingredient (Innovative Research of America #SC111). All animal groups were provided with meloxicam for the next 3 days for analgesia.

### 2.5. Horizontal Ladder Walking Assay

Locomotion performance of mice over a 10-week period following SCI was assessed via the horizontal ladder walking assay [35], as described previously [8]. Briefly, animals crossed a 1 m-long horizontal ladder consisted of sidewalls made of clear plexiglass connected by aluminum rods (ø2 mm diameter) spaced approximately 1 cm from each other. The ladder was elevated above the ground at the height of two cages. Animals were trained to cross the ladder from a neutral cage to their home cage and always in the same direction.

For handling and acclimation, the week prior to surgeries, mice settled down in the animal facility to become familiarized. For 6 consecutive days prior to testing, animals were handled for 5 min by the examiner and walked through the ladder for 5 min in order to reduce their stress during the testing procedure. Each animal was tested five times per session. Before each session, mice crossed the ladder twice for habituation. For locomotion analysis, each trial was video recorded. Videos of walking mice were analyzed frame-by-frame by two researchers blind to the condition (group, week) of the animal analyzed to identify locomotion errors, defined as steps where a hind limb missed or slipped off or was misplaced on a rod. Each fault was classified using a 7-category scale (0: total miss, 1: deep slip, 2: slight slip, 3: replacement, 4: correction, 5: partial placement, 6: correct placement) as described previously [35]. Locomotion performance was described via the fault rate (fraction of steps classified in categories 0 to 5). The last step before and the first step after a stop were excluded; only consecutive steps were included in the analysis.

While initially each animal group consisted of 9–10 mice, some of them were excluded from our final results in order to ensure consistent injury in all animals. Exclusion criteria were based on mice locomotion performance. Mice of the “Uninjured Control” group were excluded when their mean fault rate over the 10-week monitoring period exceeded 10%. Mice of the remaining four injured groups were excluded when there were strong indications of insufficient injury (e.g., fault rate <20% the day following injury or <15% at 1 wpi) or excessive injury (e.g., fault rate >90% the day following injury or >45% at 1 wpi or >35% at 2 wpi). Mice exclusions were confirmed by histology based on the magnitude of the observed lesion in mice spinal cord sections.

### 2.6. Histological Evaluation

At 12 wpi, mice were deeply anesthetized using 2.5:1 isoflurane:oxygen mix and transcardially perfused with ice-cold heparinized (10 U/mL) saline followed by 4% paraformaldehyde (PFA) in PBS. Spinal cords were dissected, post-fixed in 4% PFA at 4 °C for 1 h, washed in PBS, and immersed in 30% sucrose solution in 0.1 M phosphate buffer at 4 °C for 24 h. The 1 cm-long part of the spinal cord tissue centered around the lesion was frozen in isopentane at −70 °C and 20 μm-thick parasagittal sections were cut in super-frost slides.

For histological analysis, slides were placed in ice-cold acetone for 5 min, air-dried for 10 min in laminar flow, washed twice in PBS, once in 0.1% Triton X-100 in PBS (PBST) and once in 0.3% PBST, then blocked in 0.1% PBST supplemented with 0.1% bovine serum albumin (BSA) and 10% goat or horse serum at room temperature for 1 h, incubated in primary antibodies (L1: 1:000 rabbit polyclonal antibody (obtained by Dr. Fritz G. Rathjen)); tubulin β3 (Tubb3): 1:1000 Biolegend #801201 (clone Tuj1); glial fibrillary acidic protein (GFAP): 1:2000 Millipore #AB5541; synaptophysin: 1:200 Millipore #MAB368; green fluorescent protein (GFP): 1:200 Minotech #701-1; vesicular glutamate transporter 1 (VGLUT1): 1:400 Synaptic systems #135304; tyrosine hydroxylase (3TH): 1:40 Thermo #MA1-24654; glutamic acid decarboxylase 67 (GAD67): 1:1000 Sigma #MAB5406; α-smooth muscle actin (αSMA): 1:200 Sigma #A5228) diluted in blocking solution at 4 °C overnight, washed twice in 0.1% PBST, incubated with fluorophore-conjugated secondary antibodies (Thermo) diluted 1:1000 in PBS at room temperature for 1 h, washed twice in PBS, counterstained with Hoechst 33,342 and mounted.

Immunostained sections were imaged in a TCS SP8 inverted confocal microscope (Leica Microsystems, Wetzlar, Germany) using a 40× or a 63× oil-immersion objective lens. Images were taken at the epicenter, caudally and rostrally to the lesion. The presence of neurons was quantified by counting the density of Tuj1^+^ cells in the lesion epicenter. Astrogliosis was quantified by calculating the fraction of pixels that stained positively for GFAP immediately around the lesion boundary, by selecting up to 5 regions (approximately 0.025 mm^2^ each) per section. The presence of implanted cells was quantified by counting the density of GFP^+^ cells in the lesion epicenter, expressed per volume, using images acquired by a 63× objective lens. Synapse formation was quantified by calculating the fraction of pixels that stained positively for synaptophysin in the lesion epicenter. Axonal elongation was quantified by calculating the fraction of pixels that stained positively for L1 in the lesion epicenter as well as in dorsal region rostrally and caudally to the lesion. Angiogenesis was quantified by counting the number of αSMA^+^ vessels in the lesion epicenter. All imaging analysis was performed in Fiji software. In thresholding operations, care was taken to avoid counting the background of each staining and define positive staining using a common value chosen by an expert user.

### 2.7. Statistical Analysis

Experimental data are expressed as mean ± standard error of the mean (SEM). Statistical analysis was performed using the Prism software (Graphpad, La Jolla, CA, USA). Statistically significant effects on locomotion fault rate were assessed by 1-way analysis of variance (ANOVA; for evaluating the overall effects of injury among the groups at postop and 1 wpi) and by 2-way ANOVA (for evaluating the overall effects of BNN27 and NSC-seeded PCS graft treatments) followed by Tukey’s post hoc test (for pairwise comparisons between groups at 2–10 wpi). Statistically significant effects in the quantification of GFAP^+^, synaptophysin^+^, L1^+^ pixel fractions, Tuj1^+^ cell density and αSMA^+^ vessel number were assessed by 2-way ANOVA (for evaluating the effects of BNN27 and NSC-seeded PCS graft treatments) followed by Tukey’s post hoc test (for pairwise comparisons between groups). Statistically significant effects of BNN27 administration on the density and neuronal fate of implanted NSCs were assessed by unpaired two-tailed Student’s *t*-tests. All statistical tests assumed a statistical significance level of 0.05.

## 3. Results

Therapeutic effects of BNN27 after SCI were evaluated using the well-described mouse dorsal column crush SCI model [8]. Immediately after crush, three types of SCI treatments were tested: the first animal group was provided systemic administration of BNN27 via subcutaneous pellet (“BNN27” group; 18mg/pellet released over a 60-day period); the second animal group (“Scaffold NSC + BNN27”) was treated with a combinatorial approach consisting of a PCS implant seeded with mouse embryonic E13.5 NSCs grafted inside the lesion site immediately after injury (same graft as the one described in [8]) concurrent with the systemic delivery of BNN27 via subcutaneous pellet; the third animal group (“Scaffold NSC + Placebo”) was treated with NSC-seeded PCS implant concurrent with subcutaneous pellet that lacked an active pharmaceutical ingredient. In addition, experimental design included the “Placebo” animal group (injured mice treated with a subcutaneous pellet that lacked an active pharmaceutical ingredient) and the “Uninjured Control” animal group (where, following laminectomy, the exposed spinal cord was not injured). Grafts in the “Scaffold NSC” and “Scaffold NSC + BNN27” groups utilized NSCs isolated from mice obtained by crossing Rosa26-YFP mice containing a floxed YFP gene with CMV-Cre mice and keeping embryos whose NSCs express YFP, enabling their detection via live cell fluorescence imaging or immunocyto/histochemistry (Appendix A: Appendix A).

### 3.1. BNN27 and NSC Graft Treatment Effects on Locomotion Recovery after SCI

Mice locomotion recovery over a period of 10 weeks following dorsal column crush was evaluated by quantifying their step fault rate in the horizontal ladder walking assay (Figure 1 and Appendix A: Appendix A). During the 10-week period, mice of the “Uninjured Control” group had a time-invariant fault rate (μ = 6.9%, σ = 2.4%). No difference was observed among the four injured groups till the 1^st^ wpi, confirming that injury was consistently performed in all animals (post-op: P_1-way-ANOVA_ = 0.94, F = 0.12; week 1: P_1-way-ANOVA_ = 0.57, F = 0.69). Between 1 and 10 wpi, all injured groups showed some locomotion improvement. Eventually, differences in fault rate became statistically significant only in mice treated with NSC-seeded PCS grafts compared to the untreated injured group at 9 wpi (“Placebo”: 20.9 ± 0.5%, n = 4 vs. “Scaffold NSC + Placebo”: 13.6 ± 1.7%, n = 5, *p* = 0.01; “Placebo” vs. “Scaffold NSC + BNN27”: 14.8 ± 0.7%, n = 5, *p* = 0.04) and 10 wpi (“Placebo”: 22.8 ± 1.5%, n = 4 vs. “Scaffold NSC + Placebo”: 15.6 ± 1.2%, n = 5, *p* = 0.04; “Placebo” vs. “Scaffold NSC + BNN27”: 14.8 ± 2%, n = 5, *p* = 0.02), replicating the findings of our previous study [8]. While grafting mice with NSC-seeded PCS statistically improved locomotion recovery after 8 weeks (week 9: P_2-way-ANOVA =_ 10^−3^; week 10: P_2-way-ANOVA_ = 10^−3^), BNN27 administration did not (week 9: P_2-way-ANOVA_ = 0.89; week 10: P_2-way-ANOVA_ = 0.45). The fault rate in the “BNN27” group was not statistically different from the one of the untreated “Placebo” group (9 weeks: “BNN27”: 19.3 ± 2%, n = 4, *p* = 0.86; 10 weeks: “BNN27”: 21.1 ± 1.7%, n = 4, *p* = 0.9). Similarly, BNN27 administration did not further improve locomotion recovery when combined with NSC-seeded PCS grafts since the fault rate in the two mice groups treated with NSC-seeded PCS grafts (“Scaffold NSC + Placebo”, “Scaffold NSC + BNN27”) was not statistically different over the 10-week period (week 9: *p* = 0.91; week 10: *p* = 0.98).

### 3.2. BNN27 and NSC Graft Treatment Effects on Neuron Presence and Astrogliosis at the Lesion Site

The presence of neurons in the lesion epicenter at 12 wpi was evaluated by immunostaining spinal cord parasagittal sections for the Tuj1 neuronal marker and calculating the density of Tuj1^+^ cells within the lesion epicenter, which was denoted by a GFAP^+^ boundary region (Figure 2a). While Tuj1^+^ cells and neurites were present in all four injured groups (Figure 2b), statistical analysis revealed that both NSC-seeded PCS (P_2-way-ANOVA_ = 5∙10^−4^) and BNN27 (P_2-way-ANOVA_ = 0.01) had a statistically significant effect on the density of Tuj1^+^ cells in the lesion epicenter at 12 wpi. A significant increase in Tuj1^+^ cell density was observed in the presence of BNN27 administration or NSC-seeded PCS grafts with or without BNN27 compared to the untreated “Placebo” group (“Placebo”: 397.1 ± 53.2 cells/mm^2^, n = 4; “BNN27”: 683.8 ± 44.8 cells/mm^2^, n = 3, *p* = 0.01; “Scaffold NSC”: 776.6 ± 47.2 cells/mm^2^, n = 5, *p* = 6∙10^−4^; “Scaffold NSC + BNN27”: 789.2 ± 50.6 cells/mm^2^, n = 4, *p* = 7∙10^−4^) (Figure 2d). Simultaneous treatment with BNN27 and NSC-seeded PCS grafts did not have statistically significant synergistic effects on the density of Tuj1^+^ cells compared to treatment only with NSC-seeded PCS grafts.

Astrogliosis in the SCI lesion site at 12 wpi was evaluated by immunostaining spinal cord parasagittal sections for GFAP (marker of astrocytes) and quantifying the fraction of GFAP^+^ pixels in the lesion boundary region (Figure 2a,c). Two-way ANOVA revealed that both NSC-seeded PCS (P_2-way-ANOVA_ = 4∙10^−3^) and BNN27 (P_2-way-ANOVA_ = 0.04) had a statistically significant effect on the fraction of GFAP^+^ pixels in the lesion boundary region at 12 wpi. Compared to the untreated “Placebo” group, a significant decrease in GFAP^+^ pixel fraction was observed in the presence of BNN27 administration or NSC-seeded PCS grafts with or without BNN27 (“Placebo”: 21.1 ± 2.8%, n = 4; “BNN27”: 12.6 ± 1.5%, n = 4, *p* = 0.05; “Scaffold NSC”: 10.0 ± 0.6%, n = 3, *p* = 0.02; “Scaffold NSC + BNN27”: 8.7 ± 2.0%, n = 4, *p* = 5∙10^−3^) (Figure 2e). BNN27 administration did not lead to statistically significant synergistic effects when combined with NSC-seeded PCS grafts (“Scaffold NSC + Placebo” vs. “Scaffold NSC + BNN27”: *p* = 0.97).

### 3.3. Treatment Effects on Synapse Formation, Axonal Elongation and Angiogenesis at the Lesion Site

The integration of neurons within the lesion area via synapse formation at 12 wpi was evaluated by immunostaining spinal cord parasagittal sections for synaptophysin and then quantifying the fraction of synaptophysin^+^ pixels within the lesion epicenter (Figure 3a–c). Overall, treatment with NSC-seeded PCS had a statistically significant effect (P_2-way-ANOVA_ < 10^−4^) on the fraction of synaptophysin^+^ pixels in the lesion. Both graft-treated groups contained a larger fraction of synaptophysin^+^ pixels than the untreated “Placebo” group (“Placebo”: 4.1 ± 0.7%, n = 3; “Scaffold NSC + Placebo”: 8.4 ± 0.7%, n = 3, *p* = 2∙10^−3^; “Scaffold NSC + BNN27”: 8.7 ± 0.4%, n = 3, *p* = 10^−3^). On the other hand, BNN27 administration did not increase the fraction of synaptophysin^+^ pixels (P_2-way-ANOVA_ = 0.37; “Placebo” vs. “BNN27”: 4.8 ± 0.2%, n = 3, *p* = 0.79; “Scaffold NSC + Placebo” vs. “Scaffold NSC + BNN27”: *p* = 0.98) (Figure 3d).

Ongoing axonal elongation in the lesion site was evaluated by immunostaining spinal cord parasagittal sections for the L1 cell adhesion molecule that is upregulated in elongating axons [36] and quantifying the fraction of L1^+^ pixels in the lesion epicenter and in the dorsal regions rostrally and caudally to the lesion (Appendix A: Appendix A). No difference was observed among the four injured groups in the rostral region and in the lesion epicenter. Treatment with NSC-seeded PCS grafts had a statistically significant effect (P_2-way-ANOVA_ < 10^−4^) on the fraction of L1^+^ pixels caudally to the lesion (Appendix A: Appendix A). The two PCS graft-treated groups showed increased axonal elongation in the caudal region compared to the untreated “Placebo” group (caudally: “Placebo”: 7.1 ± 0.8%, n = 5; “Scaffold NSC + Placebo”: 11.8 ± 0.7%, n = 4, *p* < 10^−3^; “Scaffold NSC + BNN27”: 11 ± 0.4%, n = 5, *p* = 2∙10^−3^). BNN27 administration did not affect axonal elongation (caudally: P_2-way-ANOVA_ = 0.99). The fraction of L1^+^ pixels was not different in the “BNN27” group compared to the untreated “Placebo” group (caudally: “BNN27”: 7.9 ± 0.5%, n = 4, *p* = 0.83). Similarly, the combinatorial treatment with NSC-seeded PCS grafts and BNN27 administration did not increase the fraction of L1^+^ pixels compared to animals treated only with grafts (caudally: “Scaffold NSC + Placebo” vs. “Scaffold NSC + BNN27” *p* = 0.83) (Appendix A: Appendix A).

The presence of blood vessels within the lesion site was evaluated by immunostaining spinal cord parasagittal sections for αSMA, a marker of vessel-associated pericytes, and quantifying the number of αSMA^+^ vessels within the lesion epicenter (Figure 3e,f). The results show that treatment with NSC-seeded PCS grafts (P_2-way-ANOVA_ = 3∙10^−3^) but not BNN27 administration (P_2-way-ANOVA_ = 0.44) had a statistically significant effect on the number of αSMA^+^ vessels. The two PCS graft-treated groups contained an increased number of vessels within the lesion epicenter compared to the untreated “Placebo” group (“Placebo”: 2.7 ± 0.6%, n = 4; “Scaffold NSC + Placebo”: 7.2 ± 1.0%, n = 4, *p* = 0.03; “Scaffold NSC + BNN27”: 7.7 ± 0.9%, n = 3, *p* = 0.03). The number of αSMA^+^ vessels was not different in the “BNN27” group compared to the untreated “Placebo” group (“BNN27”: 4.0 ± 1.5%, n = 3, *p* = 0.82). Similarly, combinatorial treatment with NSC-seeded PCS grafts and BNN27 did not increase the number of αSMA^+^ vessels compared to animals treated only with grafts (“Scaffold NSC + Placebo” vs. “Scaffold NSC + BNN27” *p* = 0.99) (Figure 3g).

### 3.4. BNN27 Effects on Implanted NSCs

In our NSC-seeded PCS grafts, we utilized YFP-expressing NSCs so that they could be distinguished from endogenous cells by immunostaining for GFP (Figure 4 and Figure 5a). Immunostaining spinal cord parasagittal sections for GFP (implanted NSC-derived cells), Tuj1 (neurons) and GFAP (astrocytes) revealed an interesting specific spatial pattern on the fate of implanted NSC-derived cells. While Tuj1^+^GFP^+^ cells (NSC-derived neurons) remained mostly within the lesion epicenter (Figure 4a), GFAP^+^GFP^+^ cells (NSC-derived astrocytes) were localized in the lesion boundary region where they contributed to the formation of glial scar (Figure 4a,b). Double immunostaining for GFP and various neuronal markers revealed that neurons derived from implanted NSCs included inhibitory GAD67^+^ GABAergic neurons, 3TH^+^ dopaminergic neurons and VGLUT1^+^ excitatory glutamatergic neurons (Figure 4c). Interestingly, some GFP^+^ cells were detected approximately 250 μm outside the lesion, having penetrated the surrounding glial scar (Figure 4d).

BNN27 effects on the density of implanted NSC-derived cells in the lesion epicenter at 12 wpi were evaluated by counting the number of GFP^+^ cells (derived from implanted NSCs) in the two animal groups grafted with NSC-seeded PCS grafts. The results show that BNN27 administration increased the density of GFP^+^ cells in the lesion epicenter (“Scaffold NSC + Placebo”: 7.51∙10^5^ ± 1.25∙10^5^ cells/mm^3^, n = 5 vs. “Scaffold NSC + BNN27”: 15.24∙10^5^ ± 2.57∙10^5^ cells/mm^3^, n = 3, *p* = 0.02) (Figure 5a,b). In order to evaluate the effects of BNN27 administration on the neuronal fate of implanted NSCs, we counted Tuj1^+^ GFP^+^ cells in the lesion epicenter (Figure 5c). BNN27 did not affect the fraction of GFP^+^ cells that had differentiated to Tuj1^+^ neurons (“Scaffold NSC + Placebo”: 22.6 ± 5.4%, n = 4 vs. “Scaffold NSC + BNN27”: 22.0 ± 1.7%, n = 3, *p* = 0.93) (Figure 5d). However, BNN27 administration increased the fraction of Tuj1^+^ neurons that originated from implanted NSCs (“Scaffold NSC + Placebo”: 13.1 ± 3.3%, n = 4 vs. “Scaffold NSC + BNN27”: 24.4 ± 1.6%, n = 3, *p* = 0.04) (Figure 5e).

## 4. Discussion

MNTs are considered candidate therapeutic treatments for various neurologic diseases, since they mimic the effects of endogenous neurotrophins, do not cause hyperalgesia and are BBB-permeable, in contrast to endogenous neurotrophins [25]. Despite promising results in several mice neurodegeneration models, so far the therapeutic activity of MNTs had not been evaluated in any CNS injury model. Herein, we build upon our recent work on the effects of NSC-seeded PCS grafts on the dorsal column crush SCI in mice [8] and focus on the effects of BNN27 in SCI either as monotherapy or combined with NSC-seeded PCS grafts. BNN27 was delivered systemically using subcutaneous pellets over a period of approximately 60 days. Subcutaneous systemic administration is expected to provide a relatively steady BNN27 dose with no animal distraction, appropriate for minimizing artifacts in locomotion evaluation. The permeability of BNN27 in rodent CNS was demonstrated in pharmacokinetic studies where BNN27 was detected in rodent brain and retina 30 min after intraperitoneal injection [37,38].

Evaluation of locomotion recovery via the horizontal ladder walking assay showed that grafting SCI lesions with NSC-seeded PCS grafts led to statistically significant locomotion improvement after dorsal column crush in mice starting on week 9, replicating the results of our previous study [8] and highlighting the central role of NSC-based cell therapies in SCI treatments. On the other hand, systemic administration of BNN27 did not improve locomotion recovery, in agreement with a previous study where local delivery of NGF in rat SCI lesions via genetically modified fibroblasts did not lead to functional improvement 3 months post-injury [15]. Nevertheless, our histological data provide evidence that BNN27 affected several processes related to SCI wound healing, which are discussed in the following paragraphs.

Following SCI, the formation of glial scar isolates the lesion site to prevent damage spread, yet acts as a major physical barrier that obstructs axonal elongation through the lesion [2]. One way to block the inhibitory effects of a glial scar is to target its extracellular growth inhibitory components. For instance, local administration of chondroitinase ABC (ChABC), a bacterial enzyme, enhances axonal growth in CNS lesion sites by removing inhibitory chondroitin sulfate chains [39,40]. However, the clinical use of ChABC is limited by the safety of its local administration in the lesion site [41]. An alternative approach is to target astrogliosis, the accumulation and activation of astrocytes, thus impeding the major cellular component of the glial scar. Grafting SCI lesions with appropriate biomaterials, such as the PCS utilized in this study, have been reported able to reduce astrogliosis [42,43]. Here, we show that the systemic administration of BNN27 also significantly decreased astrogliosis 12 weeks following SCI (Figure 2a,c,e). This finding agrees with the report that local delivery of NGF (via genetically modified NSCs) in rat SCI lesions reduced astrogliosis [19]. While several studies have demonstrated the ability of BNN27 to reduce astrogliosis in animal models of CNS degeneration (including cuprizone-induced demyelination in mice [30], the 5xFAD Alzheimer’s disease mouse model (Karali et al. in preparation), diabetes-induced retinal damage in rats [28]), here we demonstrate for the first time the anti-astrogliotic effects of BNN27 in an animal model of CNS injury. Combinatory treatment with NSC-seeded grafts and systemic BNN27 administration further decreased the fraction of GFAP^+^ pixels compared to the animal groups that received each treatment separately, yet in a non-statistically significant manner (Figure 2e).

Emerging treatments for SCI aim to replace or reduce the loss of neural tissue induced by primary or secondary injury. In this direction, we report that grafting the SCI site with NSC-seeded PCS grafts increased neuron density in the lesion epicenter 12 weeks post SCI (Figure 2a,b,d). In the two groups where mice were grafted with NSC-seeded PCS, the increased neuron density compared to the untreated control could be attributed to multiple possible mechanisms, including the secretion of neurotrophic factors by NSCs [44] and the partial differentiation of NSCs into neurons. Quantification of Tuj1^+^ neurons in the lesion epicenter shows that 13–25% of neurons were derived from implanted NSCs (Figure 5e), suggesting that neuronal NSC differentiation cannot solely explain the increased neuronal density in the two graft-treated groups. Furthermore, we report that systemic administration of BNN27 increased neuron density in the lesion epicenter at 12 wpi (Figure 2a,b,d). This observation agrees with BNN27 neuroprotection effects reported in various neuron types in several in vitro and in vivo studies [26,27]. It also agrees with published reports on the ability of NGF administration to decrease neuron loss after SCI [16,17]. As previously reported for NGF [9,22], BNN27 effects on neurons following SCI could be mediated either directly (via NGF receptors) or indirectly via effects on other cell types. The combinatorial treatment of NSC-seeded PCS grafts and systemic BNN27 administration did not statistically increase neuronal density compared to the one in the two animal groups that received each treatment separately (Figure 2d), although BNN27 administration increased the fraction of Tuj1^+^ neurons in the lesion epicenter that originated from NSCs (Figure 5e).

A major motivation of emerging NSC-based therapies for SCI is to deliver a significant population of neural cell precursors in the lesion in order to complement the limited number of endogenous precursor cells and enhance the synthesis of new neural tissue and its connection with the surrounding spared tissue. However, previous studies have highlighted the low post-implantation survival of NSCs in SCI lesions as a major challenge that impedes the long-term efficacy of such NSC-based SCI treatments [45]. For instance, a study showed that just 4.6% of transplanted NSCs survived at 9 wpi [46]. Additionally, implanted NSCs had a low proliferation rate in situ [47]. Overall, NSC survival has been shown to depend strongly on the methodology and timepoint of delivery. NSC survival was lower when NSCs were transplanted in the acute phase of injury (1%) compared to the subacute one (6%) [48,49]. This deficiency could be caused by early immunoreaction events at the lesion site, which are toxic to implanted cells [50,51]. Strategies for protecting implanted NSCs from the harsh SCI lesion environment include delivering NSCs inside biomaterials (such as the PCS utilized in this study) and providing anti-apoptotic support via small molecule compounds or biologics. Indeed, herein we report that BNN27 administration doubled the density of implanted NSC-derived cells within the grafted lesion at 12 wpi from 7.51 × 10^5^ ± 1.25 × 10^5^ cells/mm^3^ (when injured animals were treated only with NSC-seeded grafts) to 15.24 × 10^5^ ± 2.57 × 10^5^ cells/mm^3^ (animals treated with both NSC-seeded scaffold grafts and systemic BNN27 administration) (Figure 5a,b). Comparison of these cell density measurements with data from previous studies is usually not straightforward because they utilized different experimental protocols (injury, cell seeding). Furthermore, very few published studies have quantified the number or density of cells derived from implanted NSC grafts within the lesion site. Nevertheless, our measurements on the density of implanted NSC-derived cells in the lesion are of the same order of magnitude as the ones reported in a study that grafted human neural stem/progenitor cells (NSPCs) within a fibrin–thrombin hydrogel [52].

Another major challenge of NSC-based SCI treatments is that NSCs differentiate mostly into glial cells and poorly towards neurons, as the lesion environment is unfavorable for neuronal differentiation [49,51,53,54,55]. Significant research has attempted to guide the fate of implanted NSCs using biomaterials and growth factors that enhance neuronal survival and differentiation [55]. Herein, our data show that, although BNN27 did not affect the differentiation of implanted NSCs, approximately 20% of implanted NSCs differentiated towards the neuronal fate by 12 wpi (Figure 5d), in agreement with our previous findings [8]. Furthermore, our data show an interesting spatial organization pattern for implanted NSC-derived cells in the presence or absence of BNN27 (Figure 4a). Implanted NSCs that differentiated into astrocytes became part of the glial scar and were localized in the lesion boundary region (Figure 4a,b). On the other hand, implanted NSCs that differentiated towards neurons were localized within the lesion epicenter, replenishing lost neurons. This spatial pattern has also been described by Lien et al. [56]. Finally, we demonstrate that implanted NSCs differentiated to various types of neurons (Figure 4c), including GABAergic neurons, dopaminergic neurons and glutamatergic neurons in the presence or absence of BNN27. The presence of implanted NSC-derived GAD67^+^ GABAergic neurons, VGLUT1^+^ glutamatergic and 3TH^+^ dopaminergic neurons has been reported in rat SCI models [43,57].

The observed BNN27 effects on the density and fate of implanted NSC-derived cells could originate in various mechanisms, including direct ΒΝΝ27 effects on NSCs, NSC-derived neural cells and intermediate precursors or indirect effects via other cells that participate in SCI wound healing. Regarding BNN27 effects on NSCs, various types of embryonic rodent NSCs have been reported to express both p75^NTR^ and TrkA NGF receptors [58,59]. Moreover, there are reports that NGF can increase proliferation in NSCs isolated from embryonic E14 rat brain or adult rat spinal cord [60,61]. BNN27 has demonstrated neuroprotective effects on various types of neurons mediated via NGF receptors [26,27], although no prior study has demonstrated direct effects on spinal cord neurons or NSC-derived neurons. Finally, BNN27 neuroprotective effects could also be mediated by indirect effects in SCI lesions. For instance, the ability of BNN27 to decrease astrogliosis, described above, could facilitate NSC survival and enhance the integration of NSC-derived neurons.

Apart from quantifying the effects of BNN27 in SCI, the present study further highlights the ability of NSC-seeded PCS grafts to enhance important SCI processes, which were not affected by BNN27 administration. First, grafting SCI lesions with NSC-seeded PCS grafts resulted in significantly larger fraction of synaptophysin^+^ pixels (Figure 3a–d), suggesting enhanced neuronal integration and communication in the lesion epicenter. Indeed, specific NSPC grafts have been shown able to organize, become synaptically active and interact with host axons [62]. Second, grafting SCI lesions with NSC-seeded PCS enhanced axonal elongation caudally to the epicenter (Appendix A: Appendix A), in agreement with previous measurements of axonal elongation at 6 and 9 wpi [8]. Several studies have shown that NSPC implantation can lead to extensive axonal regeneration of both host axons into the graft and axons of NSPC-derived neurons caudally to the lesion over long distances [5,57]. Third, our work shows that GFP^+^ implanted NSC-derived cells crossed the astrogliotic boundary and migrated a few hundred micrometers caudally to the lesion, but not much further to create ectopic colonies, a key limitation of NSC delivery in suspension [8]. Finally, our data show that NSC-seeded PCS enhanced angiogenesis, a critical process related to enhanced functional recovery, since endogenous angiogenesis following SCI is not sufficient to replace lost blood vessels [63,64]. Herein, we show that NSC-seeded PCS grafts increased the number of αSMA^+^ vessels within the lesion site (Figure 3e–g), in agreement with the reported ability of transplanted embryonic stem cell-derived neural progenitors to enhance angiogenesis in mouse SCI lesions [65,66].

## 5. Conclusions

This study presents the first demonstration of therapeutic effects of MNTs in an animal model of CNS injury. It shows that systemic administration of BNN27 had significant effects on astrogliosis and neuronal density following dorsal column SCI in mice. BNN27 administration also significantly increased the density of cells derived from implanted NSCs, possibly addressing a major challenge of emerging NSC-based cell therapies. Considering a wider perspective, our study suggests that MNTs can contribute to combinatorial treatments for CNS injuries by simultaneously enhancing cell therapies and modulating the loss of neuronal tissue. MNT contribution in such treatments could be further enhanced by optimizing MNT dosing, exploiting novel MNT designs that target specific neurotrophin receptors [67,68] or utilizing targeted methods to control their spatiotemporal delivery within CNS lesions.

## Figures and Tables

**Figure 1 biomedicines-11-01170-f001:**
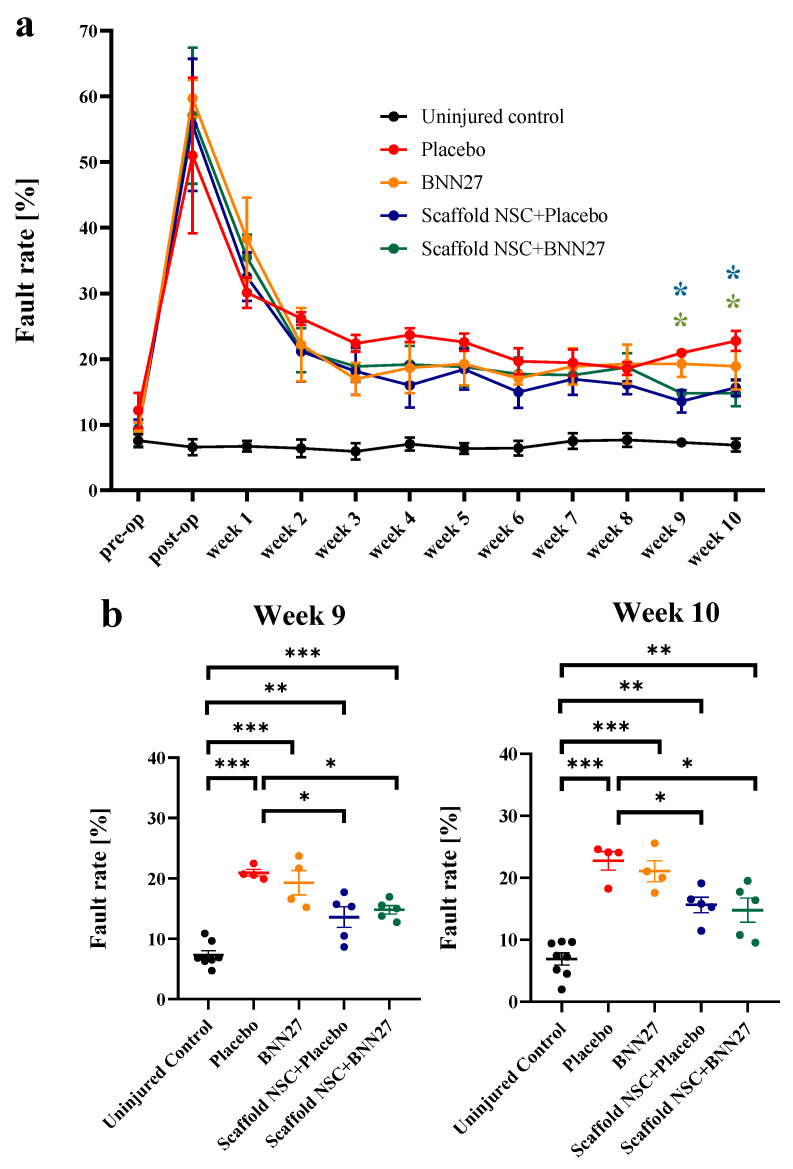
Effects of NSC-seeded PCS grafts and systemic administration of BNN27 on locomotion recovery. (**a**) Quantification of locomotion performance (fault rate) after SCI by the horizontal ladder walking assay over a period of 10 wpi (“Uninjured Control” group: n = 8, “Placebo” and “BNN27” groups: n = 4, “Scaffold NSC” and “Scaffold NSC + BNN27” groups: n = 5). The complete dataset is available in Appendix A: Appendix A. Asterisks indicate the statistically significant difference between the “Placebo” group and the “Scaffold NSC + Placebo” group (blue) or the “Scaffold NSC + BNN27” group (green) at 9 and 10 wpi. (**b**) Dot plot of locomotion fault rate at 9 and 10 wpi. Results are presented as mean ± SEM. * *p* < 0.05, ** *p* < 0.01, *** *p* < 0.001; Tukey’s post hoc pairwise test assuming P_1-way-ANOVA_ < 0.05.

**Figure 2 biomedicines-11-01170-f002:**
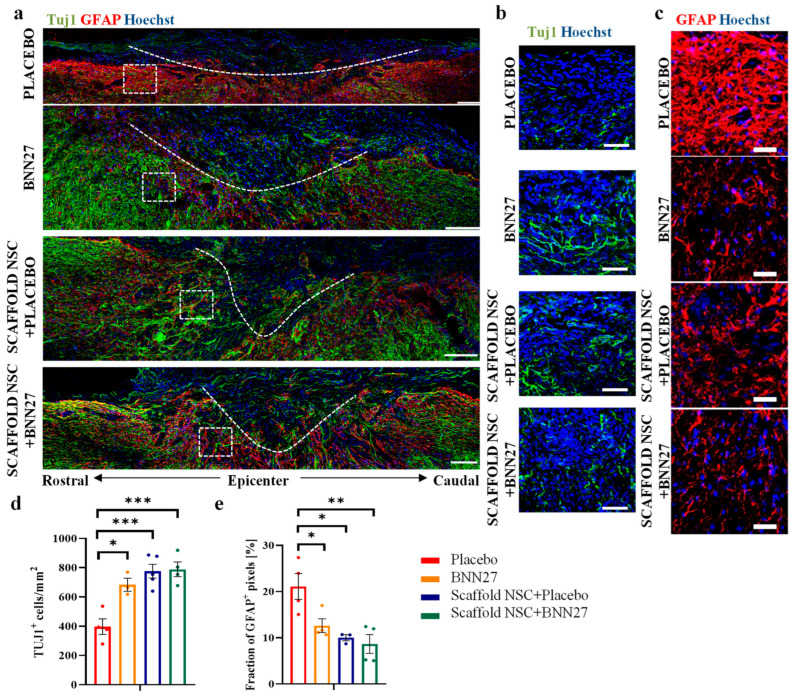
Effects of NSC-seeded PCS grafts and BNN27 administration on neuronal density and astrogliosis. (**a**) Representative fluorescence images of spinal cord parasagittal sections from all four injured groups immunostained for Tuj1 (green) and GFAP (red). The boundaries of lesion site (epicenter) are marked with a dotted line. Scale bars: 100 μm. (**b**) High-magnification fluorescence images in the lesion epicenter of image (**a**) visualizing Tuj1 immunostaining. Scale bars: 30 μm. (**c**) High-magnification fluorescence images of the rectangular ROI of image (**a**) showing GFAP immunostaining at the lesion boundary region. Scale bars: 30 μm. (**d**) Quantification of the density of Tuj1^+^ cells within the lesion epicenter at 12 wpi in the four injured animal groups (“Placebo”: n = 4, “BNN27”: n = 3, “Scaffold NSC + Placebo”: n = 5, “Scaffold NSC + BNN27” n = 4). (**e**) Quantification of the fraction of GFAP^+^ pixels (marker of astrogliosis) in the lesion boundary region at 12 wpi in the four injured animal groups (“Placebo”: n = 4, “BNN27”: n = 4, “Scaffold NSC + Placebo”: n = 3, “Scaffold NSC + BNN27” n = 4). Results are presented as mean ± SEM. * *p* < 0.05, ** *p* < 0.01, *** *p* < 0.001; Tukey’s post hoc pairwise test assuming P_2-way-ANOVA_ < 0.05.

**Figure 3 biomedicines-11-01170-f003:**
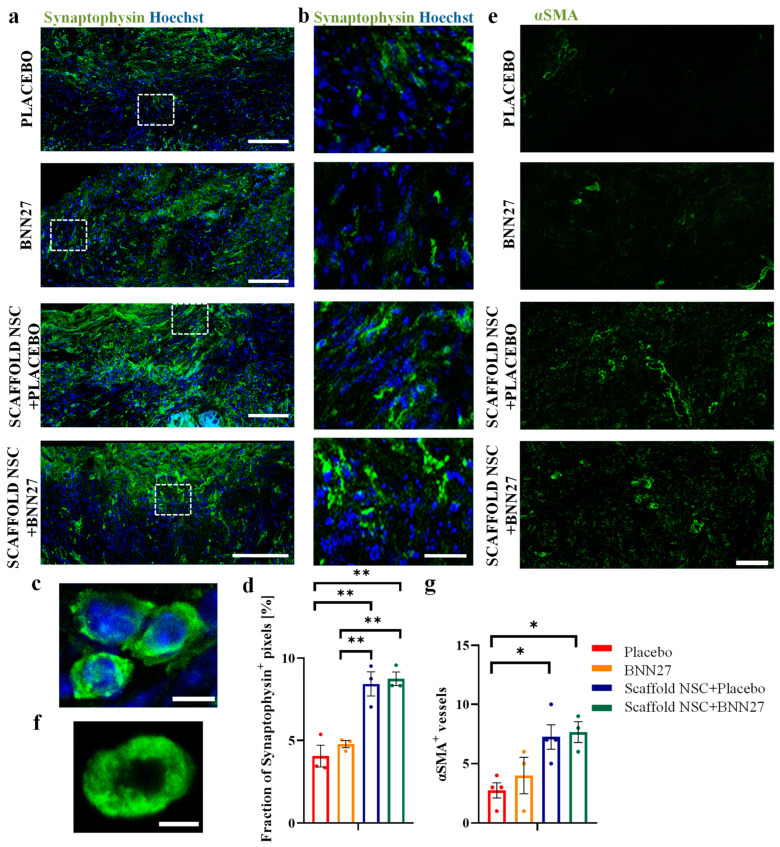
Effects of NSC-seeded PCS grafts on synapse density and angiogenesis. (**a**) Representative fluorescence images of spinal cord parasagittal sections in the lesion epicenter from all four injured groups immunostained for synaptophysin (green). Scale bars: 100 μm. (**b**) High-magnification fluorescence images of the rectangular ROIs of image (**a**) showing synaptophysin immunostaining in the lesion epicenter. Scale bars: 50 μm. (**c**) A representative high-magnification fluorescence image immunostained for synaptophysin. Scale bar: 5 μm. (**d**) Quantification of synaptophysin^+^ pixel fraction within the lesion epicenter in the four injured groups at 12 wpi (All groups: n = 3). (**e**) Representative fluorescence images of spinal cord parasagittal sections in the lesion epicenter from all four injured groups immunostained for αSMA (green). Scale bar: 50 μm. (**f**) A representative high-magnification fluorescence image of a vessel in the lesion epicenter immunostained for αSMA. Scale bar: 5 μm (**g**) Quantification of the number of αSMA^+^ vessels within the lesion epicenter in the four injured animal groups at 12 wpi (“Placebo”: n = 4, “BNN27”: n = 3, “Scaffold NSC + Placebo”: n = 4, “Scaffold NSC + BNN27”: n = 3). Results are presented as mean ± SEM. * *p* < 0.05, ** *p* < 0.01; Tukey’s post hoc pairwise test assuming P_2-way-ANOVA_ < 0.05.

**Figure 4 biomedicines-11-01170-f004:**
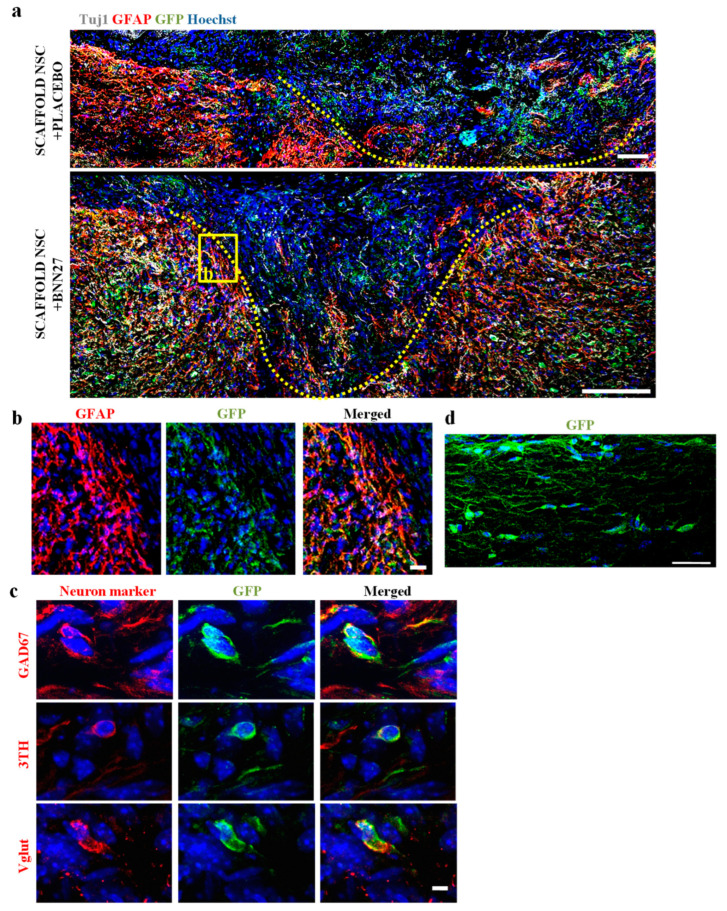
Spatial patterning of implanted NSC fate at the spinal cord lesion site. (**a**) Representative fluorescence images of spinal cord parasagittal sections of SCI lesion immunostained for Tuj1^+^ neurons (gray), GFAP^+^ astrocytes (red) and GFP^+^ implanted NSC-derived cells (green). The boundaries of lesion sites are marked with a dotted line. Scale bars: 100 μm. (**b**) High-magnification fluorescence image of the rectangular insert of image (**a**) showing GFP^+^GFAP^+^ implanted NSC-derived astrocytes at the lesion boundary region. Scale bar: 30 μm. (**c**) Representative high-magnification fluorescence images of GFP^+^ implanted NSC-derived cells double-immunostained with markers of different neuron types: GAD67 (inhibitory GABAergic neurons; upper row), 3TH (dopaminergic neurons; middle row) and VGLUT1 (excitatory glutamatergic neurons; lower row) (red). Scale bar: 3μm. (**d**) Representative fluorescence image of a spinal cord parasagittal section showing GFP^+^ implanted NSC-derived cells located in the dorsal column, approximately 250 μm caudally to the lesion. Scale bar: 30 μm.

**Figure 5 biomedicines-11-01170-f005:**
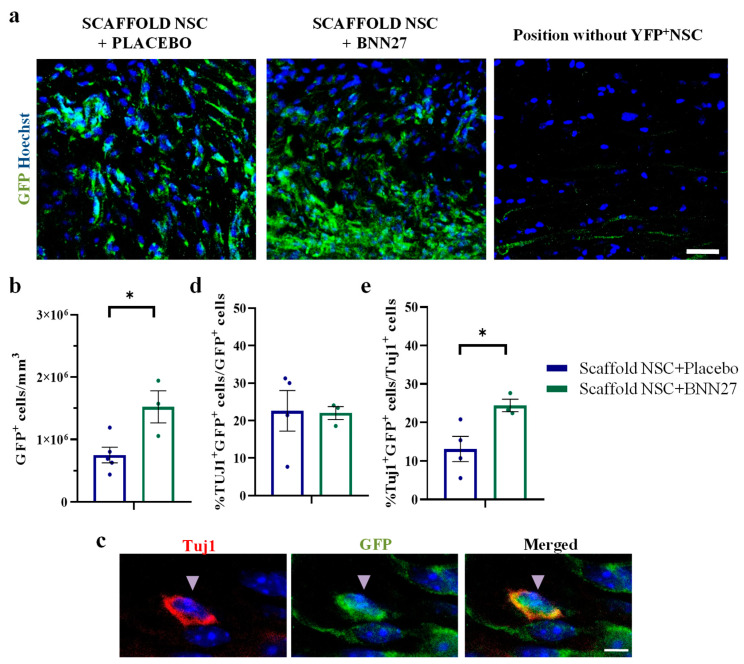
Effects of BNN27 administration on the density and neuronal fate of implanted NSCs. (**a**) Representative high-magnification fluorescence images of GFP^+^ implanted NSC-derived cells from the two animal groups (“Scaffold NSC + Placebo”, “Scaffold NSC + BNN27”) treated with NSC-seeded PCS and a region without implanted NSCs. Scale bar: 30 μm. (**b**) Quantification of the density of GFP^+^ cells in the lesion epicenter at 12 wpi. (**c**) High magnification fluorescence images of cells within the lesion epicenter double-stained for GFP (green; implanted NSC-derived cells) and Tuj1 (red; neurons). Scale bars: 5 μm. (**d**) Quantification of the fraction of GFP^+^ cells that also stain for the Tuj1 in the lesion epicenter at 12 wpi. (**e**) Quantification of the fraction of Tuj1^+^ neurons that also stain for GFP in the lesion epicenter at 12 wpi (“Scaffold NSC + Placebo”: n = 4, “Scaffold NSC + BNN27”: n = 3). Results are presented as mean ± SEM. * *p* < 0.05; unpaired two-tailed Student’s *t*-test.

## Data Availability

Data are available within the article or in Appendix A. Raw data are available upon reasonable request.

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
