# Peer review of "Microneurotrophin BNN27 Reduces Astrogliosis and Increases Density of Neurons and Implanted Neural Stem Cell-Derived Cells after Spinal Cord Injury"

_biomedicines, 2023, doi:10.3390/biomedicines11041170_

Round 1

Reviewer 1 Report

In the manuscript entitled ‘Microneurotrophin BNN27 reduces astrogliosis and increases the density of neurons and implanted neural stem cell-derived cells after spinal cord injury’, authors have investigated the therapeutic efficacy of BNN27 in combination with combined with neural stem cell (NSC)-seeded collagen-based scaffold grafts in the mouse dorsal column crush spinal cord injury (SCI) model. They have reported significant post-treatment recovery with increased neuronal population and reduced astrogliosis at the site of the incision. We still have some concerns to be addressed.

1.       In the locomotion recovery test (figure 1), there is no difference between “Scaffold NSC + placebo” and “Scaffold NSC+BNN27”. Is only Scaffold NSC itself sufficient for improvement in locomotion? We also do not see the difference between “Placebo” and “BNN27”. Can you explain?

2.       In figure 2, representative images (2a, b) do not match the quantification of Tuj-positive and GFAP-positive cells. Can you add a magnified image of GFAP and indicate the magnified regions in fig 2a?

3.       Just to nullify the background immunostaining, I would suggest you add staining of non-neuronal tissues in supplementary for the images in figure-2 and 3.

4.       Were GFP-expressing NSCs transplanted in all the treatment groups?

5.       How do you explain the effect of BNN27 in implantation studies which seems ineffective?

Reviewer 2 Report

This is a very interesting study concerning the effect of systematic administration of microneurotrophins (BNN27) to enhance the transplanted neuronal stem cells proliferation and inhibit the astrogliosis which contributed to the improvement in neurological function in the spinal cord injury animal model. The study was well designed and written. But, in the well development of co-culture system, there was lack of data in increased migration in neuronal stem cells and inhibited migration in astrocytes exerted by the BNN27. Either the authors added the above said experiment, or cited the related reference with the data of BNN27 on the above said effect.

Author Response

This study presents the first investigation on MNT effects in an animal model of SCI. Indeed, the systemic administration of BNN27 did not result in locomotion improvement. Yet results show some specific BNN27 effects on astrogliosis and neuronal loss in SCI lesions. Furthermore, when BNN27 administration was combined with the implantation of NSC-seeded scaffolds, BNN27 increased the density of NSC-derived cells.

Our data show a particular spatial pattern of NSC-derived astrocytes and NSC-derived neurons within the SCI lesion (fig. 4). Furthermore, we observed the presence of YFP+ (NSC-derived) cells a few hundred μm caudally to the lesion (fig. 4). These results probably arise from a combination of NSC proliferation, migration and differentiation. We indeed did not study the migration of NSC-derived cells in vivo, as this would require elegant imaging techniques not available to us. Nevertheless, a time-lapsed imaging study of implanted NSCs within a SCI lesion could be an exciting future research direction.

Finally, we did not complement our in vivo animal studies with in vitro studies of BNN27 effects on NSCs. This choice was based on the lack of in vitro systems able to recapitulate the complex microenvironment of a SCI lesion. As mentioned in the discussion, BNN27 effects on NSCs in SCI lesions could be indirect, that is mediated via other cell types (in contrast to direct effects on NSC receptors). Studies of such indirect effects could indeed be implemented using emerging in vitro systems that enable co-cultures of NSCs with other cell types present in SCI lesions direction.

Round 2

Reviewer 1 Report

I appreciate the authors for their effort made to improve the manuscript.